# Contrariety of Human Bone Marrow Mesenchymal Stromal Cell Functionality in Modulating Circulatory Myeloid and Plasmacytoid Dendritic Cell Subsets

**DOI:** 10.3390/biology12050725

**Published:** 2023-05-16

**Authors:** Crystal C. Uwazie, Tyler U. Faircloth, Rhett N. Parr, Yenamala U. Reddy, Peiman Hematti, Devi Rajan, Raghavan Chinnadurai

**Affiliations:** 1Department of Biomedical Sciences, Mercer University School of Medicine, Savannah, GA 31324, USA; 2Department of Medicine, Medical College of Wisconsin, Milwaukee, WI 53226, USA

**Keywords:** mesenchymal stromal cells, secretome, circulating dendritic cells, immunomodulation

## Abstract

**Simple Summary:**

Mesenchymal Stromal/Stem cells (MSCs) are non-hematopoietic cells of the bone marrow that possess immunomodulatory and regenerative properties. MSCs are widely being tested in clinical trials as a cellular therapy for inflammatory and degenerative disorders. In addition, regulatory authorities in some countries have approved MSC-based cellular therapy for certain ailments such as Graft vs. Host Disease, Crohn’s-disease-associated perianal fistula, and Critical limb ischemia. Knowledge is emerging relating to MSCs’ potency and interaction with host immune components to inform sustained therapeutical benefit. In the present study, we defined the interaction between human bone-marrow-derived MSCs and circulatory dendritic cell populations, which are the key mediators of immunomodulation. This study provides insights into MSC’s modulatory effects on circulating dendritic cell subsets and the associated secretome signature, which could predict MSC’s potency and biomarker evaluation.

**Abstract:**

Mesenchymal Stromal Cells (MSCs) derived from bone marrow are widely tested in clinical trials as a cellular therapy for potential inflammatory disorders. The mechanism of action of MSCs in mediating immune modulation is of wide interest. In the present study, we investigated the effect of human bone-marrow-derived MSCs in modulating the circulating peripheral blood dendritic cell responses through flow cytometry and multiplex secretome technology upon their coculture ex vivo. Our results demonstrated that MSCs do not significantly modulate the responses of plasmacytoid dendritic cells. However, MSCs dose-dependently promote the maturation of myeloid dendritic cells. Mechanistic analysis showed that dendritic cell licensing cues (Lipopolysaccharide and Interferon-gamma) stimulate MSCs to secret an array of dendritic cell maturation-associated secretory factors. We also identified that MSC-mediated upregulation of myeloid dendritic cell maturation is associated with the unique predictive secretome signature. Overall, the present study demonstrated the dichotomy of MSC functionality in modulating myeloid and plasmacytoid dendritic cells. This study provides clues that clinical trials need to investigate if circulating dendritic cell subsets in MSC therapy can serve as potency biomarkers.

## 1. Introduction

Mesenchymal Stromal Cells (MSCs) are non-hematopoietic stem cells present in bone marrow and several visceral tissues [1,2,3]. MSCs possess numerous regenerative and immunomodulatory properties, promoting their application as cellular therapy. The therapeutic potential of MSCs is being investigated in several clinical trials for potential inflammatory and degenerative disorders [4,5]. Importantly, MSC-based cellular therapy is already approved in some countries as a standard clinical care for certain disorders such as such as Graft-versus-Host Disease (GvHD), Perianal Fistula in Crohn’s Disease, and Critical Limb Ischemia [6,7,8]. However, concerns remain since challenges exist in accomplishing reliable efficacy [9,10,11]. The future of MSC therapy relies on the demonstration of consistent efficacy in the clinic. This is challenging since several factors, including host predisposition, dose, route and the critical quality attributes of MSCs, contribute to clinical responses [12,13]. In addition, identifying the biomarkers of predictive MSC therapy and the definitive mechanism of action are still developing [14]. Once MSCs are infused into the patients, they interact with several host immune and pathological cellular components, resulting in immunomodulation and tissue regeneration. Hence, the characterization of MSCs’ granular interaction with circulating immune effectors is necessary and can inform the potency of MSC therapy. Previous studies have extensively investigated MSCs’ interaction with lymphoid populations, such as T and B lymphocytes, and also monocytes [15,16,17,18,19]. However, the effect of MSCs on dendritic cell subsets needs further investigation.

Dendritic cells are professional antigen-presenting cells that instruct T cells, and they respond to exogenous microenvironmental cues that allow them to mature, migrate, and interact with other leukocytes that collectively shape the immune system [20,21]. Although distinct dendritic cell subsets have been identified, predominant populations, such as myeloid (CD1c+/BDCA1+) and plasmacytoid dendritic cells (CD303+/BDCA2+), possess major immunomodulatory roles [22,23,24,25]. CD1c+ myeloid dendritic cells, the largest dendritic cell subsets in human lympho-hematopoietic tissues, can also be identified in the circulating peripheral blood [26,27,28]. Plasmacytoid dendritic cells are the major producers of interferon alpha (IFNα) and also regulate immune responses like myeloid dendritic cells [29,30]. Dendritic cells normally present in an inactivated/immature state, and stimulation/licensing with exogenous cues lead to their maturation, substantially enhancing their activity. Mature dendritic cells are identified through their high expression of immunoglobulin superfamily member CD83 [31,32]. Although some studies have provided evidence that MSCs modulate the maturation of in vitro monocyte-derived dendritic cells, the current knowledge on the effect of MSCs on circulating dendritic cell subsets in the peripheral blood is lacking. Ex vivo investigation of circulating dendritic cells taken from the peripheral blood (without manufacturing dendritic cells from monocytes) will inform the clinical relevance of MSC therapy since infused MSCs encounter cellular compartments of the peripheral blood, including dendritic cells. In addition, circulating dendritic cell frequency and functionality vary in several diseases, which also indicates its significance in systemic immune modulation [33,34,35,36]. Thus, in the present study, we aim to identify the effect of human bone-marrow-derived MSCs on circulating myeloid and plamacytoid dendritic cells upon their coculture ex vivo. Our results demonstrated the dichotomy of MSC functionality in modulating myeloid and plasmacytoid dendritic cells. We identified that MSCs promote the maturation of myeloid dendritic cells. In contrast, MSCs do not significantly modulate the responses of plasmacytoid dendritic cells. Mechanistic analysis showed that dendritic cell licensing cues (LPS+IFNγ) stimulate MSCs to secret an array of dendritic cell maturation-associated secretory factors. We also identified that MSC-mediated upregulation of myeloid dendritic cell maturation is associated with the unique secretome signature. The present study provides clues that future clinical trials need to investigate circulating dendritic cell subsets in MSC therapy.

## 2. Materials and Methods

### 2.1. Human Bone Marrow MSCs

Discarded and de-identified bone marrow filters were used for the isolation of bone marrow mononuclear cells (MNCs). Bone marrow filters were collected at the end of the bone marrow harvest from healthy donors following IRB protocol #2016-0298 at the University of Wisconsin Madison. The Ficoll^TM^ (Cytiva, Marlborough, MA, USA) density gradient centrifugation process was performed to enrich the mononuclear cells (MNCs) from the bone marrow. MNCs were seeded at a density of 200,000–300,000 cells/cm^2^ in 1× alpha-Minimal Essential Medium (Corning, New York, NY, USA) containing 10% human platelet lysate (Mill Creek Life Sciences, Rochester, NY, USA), L-glutamine (Corning, New York, NY, USA), and 100 IU/mL penicillin/streptomycin/Amphotericin B (Corning, New York, NY, USA). The seeded MNCs were incubated (37 °C, 5% CO_2_, humidified incubator) for three days. Non-adherent cells were aspirated, and fresh medium was replaced three days after seeding; subsequently, the cultures were maintained with fresh media every 48–72 h until the observance of the colonies. The adherent cells were trypsinized and reseeded at a density of 5000 cells/cm^2^ in 1X alpha-Minimal Essential Medium containing 5% human platelet lysate and 100 IU/mL penicillin/streptomycin/Amphotericin B. Upon passage 2, the identities of the MSCs were confirmed through flow cytometry. MSCs were regularly passaged upon confluency of 70–80% and were cryopreserved until they were needed for future experiments. MSCs derived from the passages between two and six were used for the experiments.

### 2.2. Coculturing of MSCs and Peripheral Blood Mononuclear Cells

Human peripheral blood mononuclear cells (PBMCs) were isolated from the peripheral blood, and leukapheresis products were obtained from healthy donors. PBMCs isolated through the Ficoll gradient centrifugation process were cryopreserved until they were needed for the experiments. At the time of the experiment, PBMCs were thawed and rested in the incubator for several hours before being subjected to further procedures. Dendritic-cell-enriched PBMCs were adapted according to the EasySep™ Human Pan-DC Pre-Enrichment Kit (STEMCELL Technologies, Vancouver, Canada). For the coculture, MSCs were seeded onto a 96-well plate at appropriate cell densities with 1× alpha-Minimal Essential Medium containing 10% fetal bovine serum and 100 IU/mL penicillin/streptomycin/Amphotericin B. The cocultures were stimulated with the appropriate concentrations of R837 (20 ug/mL) (Invivogen, San Diego, CA, USA) or recombinant human IFNγ (40 ng/mL) (Thermofisher, Waltham, MA, USA) and Lipopolysaccharide (100 ng/mL) (LPS) (Invivogen, San Diego, CA, USA) in 1X RPMI Medium (Corning, New York, NY, USA) containing 10% fetal bovine serum and 100 IU/mL penicillin/streptomycin/Amphotericin B. For the intracellular cytokine capture assay with R837, Golgi transport inhibitor, BD Golgi Plug™ (BD Biosciences, Franklin Lakes, NJ, USA) was added after three hours of initial culture and were further incubated overnight until they were subjected to intracellular cytokine staining for flow cytometry. For the experiments with LPS and IFNγ, the cocultures were harvested for flow cytometry after 24 h of incubation.

### 2.3. Flow Cytometry

Dendritic-cell-enriched PBMCs were harvested from under the cell culture conditions and stained with the appropriate antibodies, Lineage Cocktail FITC (clones UCHT1; HCD14; 3G8; HIB19; 2H7; HCD56), CD1C PE Dazzle 594 (Clone L161), CD303 APC (Clone S19008G), HLADR APC/Cy7 (CloneL243), CD83 PE (Clone HB15e) (Biolegend, San Diego, CA, USA). Intracellular flow cytometry staining was performed using the BD Cytofix and Cytoperm procedure (BD Biosciences, Franklin Lakes, NJ, USA) with the antibodies IFN-α PE (Miltenyi Biotec, Gaithersburg, MD, USA). MSC identity was confirmed with the staining panel including CD45 FITC (Clone HI30), CD105 APC (Clone 266), CD44 APC (Clone G44-26), CD90 APC (Clone 5E10), CD73 PE (Clone AD2) (BD Biosciences, Franklin Lakes, NJ, USA). The stained cells were subsequently acquired in BD FACS Aria II flow cytometer. Flow cytometry plots were analyzed and generated utilizing FlowJo software. The following gating strategy was used to define dendritic cell subsets. Forward and side scatter gated populations were selected for single cells and excluding doublets. Lineage-negative populations were further gated for HLADR expression. The HLADR-positive populations were gated based on the expression of CD103 and CD1C. This gating strategy identified plasmacytoid (Lineage-(CD3-CD14-CD16-CD56-CD19-CD20-)HLADR+CD303+CD1C-) and myeloid (Lineage-(CD3-CD14-CD16-CD56-CD19-CD20-)HLADR+CD303-CD1C+) dendritic cell subtypes.

### 2.4. Multiplex Analysis

The supernatants derived from the coculture of MSCs and dendritic cells were harvested and cryostored. To identify the secretome responses of the MSCs secretome, the cells were seeded onto 96-well plates at a density of 10,000 cells per well. The next day, they were stimulated with the appropriate concentrations of LPS and IFNγ. Two days later, the supernatants were collected and cryostored at −80 °C. At the time of multiplex analysis, the supernatants were thawed from cryostorage and centrifuged at 2500 rpm for 10 min to remove debris. Immunomodulatory 30-Plex human panel magnetic bead multiplex assay (Thermofisher, Waltham, MA, USA) was performed according to the manufacturer’s instructions. Cytokines in the panel include EGF (Epidermal Growth Factor), Eotaxin, FGF-2, G-CSF (Granulocyte Colony-Stimulating Factor), GM-CSF (Granulocyte–Macrophage Colony-Stimulating Factor), HGF (Hepatocyte Growth Factor), IFNα, IFNγ, IL-10, IL-12, IL-13, IL-15, IL-17A, IL1β, IL-1RA, IL-2, IL-2R, IL-4, IL-5, IL-6, IL-7, IL-8, IP-10, MCP-1 (Monocyte Chemoattractant Protein-1), MIG (Monokine Induced By Gamma), MIP-1α (Macrophage Inflammatory Protein-1 Alpha), MIP-1β (Macrophage Inflammatory Protein-1 Beta), RANTES, TNFα, and VEGF (Vascular Endothelial Growth Factor). Luminex^TM^ xMAP (multianalyte profiling) software was then utilized to acquire and analyze the results. Concentrations of secretory molecules are quantified as picograms per milliliter (pg/mL).

### 2.5. Statistics

GraphPad Prism 9.0 software was used to analyze data. Linear regression analysis was utilized to attain *p*-values and correlation coefficients (r). r-value of 1 and −1 indicates the best direct and inverse correlations, respectively, while 0 means no correlation. The degree of statistical significance was determined based on appropriate *p*-values (ns, *p* > 0.05; * *p* ≤ 0.05; ** *p* ≤ 0.01; *** *p* ≤ 0.001; **** *p* ≤ 0.0001). Negative logarithmic *p*-values are determined to depict the *p*-values in the heatmap.

## 3. Results

### 3.1. Lipopolysaccharide (LPS) and Interferon Gamma (IFNγ) Induce the Maturation of Dendritic Cells with the Modulation of Secretome

In order to define the effect of Mesenchymal Stromal Cells (MSCs) on circulating dendritic cell subsets, we developed a feasible methodology to characterize dendritic cell responses. Considering the very low frequency of circulating dendritic cells, we enriched the dendritic cell populations in the leukapheresis-driven Peripheral Blood Mononuclear Cells (PBMCs). We utilized multicolor flow cytometry to characterize the dendritic cell subsets, which identified plasmacytoid (Lineage-(CD3-CD14-CD16-CD56-CD19-CD20-)HLADR+CD303+CD1C-) and myeloid (Lineage-(CD3-CD14-CD16-CD56-CD19-CD20-)HLADR+CD303-CD1C+) dendritic cell subtypes (Figure 1A). The stimulation of dendritic-cell-enriched PBMCs with LPS and IFNγ for 24 h leads to the maturation of dendritic cells, as observed with the upregulation of the maturation marker, CD83 (Figure 1B–E). Although CD83 is upregulated in both plasmacytoid and myeloid dendritic cell subsets, the pronounced effect is seen only in myeloid dendritic cell subsets (Figure 1D,E). We have also investigated the supernatants of the dendritic-cell-enriched PBMCs with and without stimulation for multiplex secretome analysis of cytokines and growth factors using Luminex xMAP technology (Figure 1F). Our results demonstrate that 14 out of 30 secretory molecules (IL-6, MIP-1a, TNFa, MIP-1b, IL-1b, GMCSF, IL-8, IL-10, IL-12/IL23p40, RANTES, IP-10, VEGF, GCSF, and IL-2R) are secreted upon stimulation with differential magnitude and statistical significance (Figure 1F,G). We did not include IFNγ in the analysis, as this is used for the stimulation. In contrast, one secretory molecule, MCP-1, is downregulated upon stimulation (Figure 1G).

### 3.2. Phenotypical Analysis of Human Bone-Marrow-Derived Mesenchymal Stromal Cells (MSCs)

We isolated and expanded MSCs from the bone marrow of healthy individuals since hematological malignancies or malfunctions might affect the biology of MSCs. The filters of the bone marrow harvested from six healthy individuals, who donated hematopoietic stem cells for transplantation, were used to isolate MSCs. The isolated MSCs exhibit MSC phenotypical identity as CD45−, CD105+, CD73+, CD44+, and CD90+ non-hematopoietic stem cell populations (Figure 2).

### 3.3. LPS and IFNγ Stimulation Induces an Array of Dendritic Cell Maturation-Associated Secretory Factors on Human Bone Marrow MSCs

We determined the effect of LPS and IFNγ stimulation on human bone marrow MSCs. We stimulated MSCs derived from five independent donors with escalating concentrations of LPS and IFNγ (100 + 40, 10 + 4, 1 + 0.4, 0 + 0 LPS + IFNγ ng/mL). The supernatants derived from these conditions were subjected to secretome analysis using Luminex^TM^ xMAP technology. Our results demonstrate that LPS and IFNγ stimulations substantially upregulate (difference in more than 100 pg/mL when comparing stimulation and no stimulation conditions) six secretory molecules, including IL-6, MCP-1, IL-8, IP-10, RANTES, and MIG (Figure 3). Fifteen other secretory molecules, including MIP-1b, IL-15, EGF, IL-2R, G-CSF(CSF-3), MIP-1a, IFN-a, FGF-2, GM-CSF, IL-1RA, IL-2, IL-10, and IL-17A(CTLA-8), IL-1b, Eotaxin (CCL-11), show a statistically significant upregulation, though their magnitude of upregulation is low (difference in less than 20 pg/mL when comparing stimulation and no stimulation conditions) (Figure 3). The remaining eight secretory molecules (HGF, VEGF, IL-13, TNFa, IL-7, IL-12/IL-23p40, IL-5, and IL-4) did not show any significant modulation upon LPS and IFNγ stimulation (Figure 3).

### 3.4. Human Bone-Marrow-Derived MSCs Further Promote the Maturation of Circulating Myeloid Dendritic Cell Subsets upon Coculture

To determine the effect of human MSCs in modulating circulating dendritic cell maturation, we cocultured human MSCs derived from six donors with dendritic-cell-enriched PBMCs from independent donors in various ratios. Dendritic cell populations in the PBMCs were activated with appropriate concentrations of LPS and IFNγ. Twenty-four hours post coculture, plasmacytoid (Lineage-(CD3-CD14-CD16-CD56-CD19-CD20-)HLADR+CD303+CD1C-CD83+) and myeloid (Lineage-(CD3-CD14-CD16-CD56-CD19-CD20-)HLADR+CD303-CD1C+CD83+) dendritic cell maturation was quantified using flow cytometry (Figure 4A,B). Our results demonstrate that MSCs do not modulate the maturation of plasmacytoid dendritic cells (Figure 4B,D). In contrast, the maturation of myeloid dendritic cells was increased in a dose-dependent fashion based on MSC numbers (Figure 4D). Altogether these results suggest that human bone-marrow-derived MSCs promote the maturation of circulating myeloid dendritic cells.

### 3.5. Secretome Responses of LPS and IFNγ Activated MSC and Dendritic Cells Enriched PBMC Coculture

To identify the secretory molecules of MSC and dendritic-cell-enriched PBMC coculture, we evaluated the supernatants using Luminex xMAP technology. Our results have identified three patterns of secretome modulation (Figure 5). (1) Eighteen secretory molecules are dose-dependently upregulated in the cocultures based on MSC numbers, (2) 10 secretory molecules are not modulated upon MSC introduction, (3) One cytokine, TNFa, is dose-dependently downregulated depending on MSC numbers (Figure 5). As identified earlier in Figure 1G, the 14 secretory molecules (IL-6, MIP-1a, TNFa, MIP-1b, IL-1b, GMCSF, IL-8, IL-10, IL-12/IL23p40, RANTES, IP-10, VEGF, GCSF, and IL2R) that are upregulated by LPS and IFNγ on dendritic-cell-enriched PBMC cultures (in the absence of MSCs) display three secretome patterns upon the introduction of MSCs. (A) Seven secretory molecules, MIP-1a, MIP-1b, IL-1b, GMCSF, IL-8, IL-10, and IL-12/IL23p40, are not modulated upon MSC introduction; (B) Six secretory molecules IL-6, RANTES, IP-10, VEGF, IL2R, and GCSF are further upregulated upon MSC introduction; (C) One secretory molecule, TNFa is downregulated upon MSC introduction (Figure 5). Altogether, these results suggest that MSCs predominantly do not attenuate the secretome of LPS- and IFNγ-activated dendritic-cell-enriched PBMC cocultures.

### 3.6. Secretome Signature Predicts MSC’s Interaction with Myeloid but Not Plasmacytoid Dendritic Cells

We determined the relationship between MSC’s effect on the maturation of circulating myeloid, plasmacytoid dendritic cells and secretory molecules. The percentage of plasmacytoid (Lineage-(CD3-CD14-CD16-CD56-CD19-CD20-)HLADR+CD303+CD1C-CD83+) and myeloid (Lineage-(CD3-CD14-CD16-CD56-CD19-CD20-)HLADR+CD303-CD1C+CD83+) subsets were subjected to linear regression analysis with secretory molecules. The correlation coefficient values represent the degree of correlation. r = 1 and r = −1 indicate the best direct and inverse correlation, respectively while r = 0 represent no correlation. Our results have identified that myeloid dendritic cell maturation in the presence of MSCs is directly correlated with the 19 secretory molecules with high statistical significance (Figure 6A,C). The ranking of this direct correlation includes RANTES, VEGF, IP-10, GCSF, IL-6, MIG, IL-17A, EGF, IL-7, MIP-1a, MIP-1b, IL-15, MCP-1, IL-4, IFN-a, IL-2R, Eotaxin, IL-2, IL-1b (Figure 6A,C). Plasmacytoid dendritic cell maturation is associated with only one secretory molecule, GMCSF, with high statistical significance (Figure 6B,D). These differential correlative secretory signatures reiterate the dichotomic effect of MSCs on the maturation of plasmacytoid and myeloid dendritic cells. In addition, these results further suggest that the human bone marrow MSC-mediated upregulation of myeloid dendritic cell maturation predominantly correlates with secretory molecules in a direct but not inverse manner.

### 3.7. Human Bone Marrow MSCs Do Not Block the Key Function of Plasmacytoid Dendritic Cells

Plasmacytoid dendritic cells are the major producers of IFNα during infection and inflammation. We determined the effect of MSCs in modulating the key response of plasmacytoid dendritic cells by measuring IFNα through the use of intracellular cytokine capture analysis. We have cocultured MSCs with R837 (TLR7 agonist)-stimulated PBMCs for 14 h in the presence of a Golgi transport inhibitor. R837 induces IFNα production only in plasmacytoid but not myeloid dendritic cell populations (Figure 7A). Our results with five independent donors have demonstrated that MSCs do not block IFNα secretion from plasmacytoid dendritic cells (Figure 7B,C). Altogether these results suggest that human bone marrow MSCs do not block the key function of plasmacytoid dendritic cells.

## 4. Discussion

In the present study, we demonstrated the dynamics of human bone-marrow-derived MSC’s interaction with circulating dendritic cell subsets. Our results demonstrate that MSCs do not attenuate the maturation of circulating dendritic cell subsets. Conversely, the maturation of myeloid dendritic cells is upregulated upon MSC exposure. Previous studies reported that MSCs block the maturation of monocyte-derived dendritic cells [37,38,39,40]. In those studies, dendritic cells are investigated after their in vitro differentiation from monocytes with appropriate cues through a lengthy culture procedure. However, our analysis, which solely focuses on the circulating dendritic cells of the peripheral blood ex vivo under short-term stimulation conditions, unambiguously shows that MSCs fail to inhibit the responses of plasmacytoid and myeloid dendritic cells while promoting the maturation of myeloid dendritic cells. These observations are relevant to the clinical translation of MSC-based cell therapy, where intravenously infused MSCs largely home to the lungs and captured in microcapillaries where they encounter circulating immune cell components of the lymphomyeloid and dendritic cell populations [41,42]. In fact, the short persistence of MSCs and biodistribution provided some mechanistic understanding that its therapeutic function is similar to the “hit-and-run” mode of action [43]. In supporting this, our experimental assay system is designed with short-term stimulation conditions (14 or 24 h) and provides translational insights related to MSCs’ instant interaction with the circulating responding dendritic cell populations.

Our results show MSC-mediated increment of the maturation of circulating myeloid dendritic cells, which may explain two mechanistic insights in providing the translational benefit of MSC therapy. A study reported that intravenous infusion of umbilical cord MSCs in patients with Systemic lupus erythematosus (SLE), resulting in the increased frequency of CD1c+ dendritic cells. This provided insights into the understanding of MSC therapy in modulating inflammation through the upregulation of CD1c+ dendritic cells [44]. Secondly, it has been emerging that efferocytosis, a process in which dead cells are cleared through phagocytosis, plays a role in the therapeutical function of MSCs. It has been shown that infused MSCs undergo apoptosis and are subsequently phagocytosed by the host phagocytes, which exerts immunomodulation [45,46,47]. It is also emerging that circulating migratory and resident phagocytic cells of the secondary lymphoid organs play roles in mediating the therapeutic effect of MSCs through this efferocytosis process [48]. Thus, it is entirely possible to speculate and confirm in future studies that MSCs induce the maturation of dendritic cells and subsequently undergo dendritic-cell-based efferocytosis, resulting in cumulative immune modulation. Similarly, future studies are also warranted to define the effect of MSC-induced maturation of dendritic cells on their antigen presentation and T-cell priming functions.

We have identified at least 14 secretory molecules that are secreted upon the stimulation of dendritic-cell-enriched PBMC cultures in the absence of MSCs. Out of these fourteen secretory molecules, seven are not modulated, and six are further upregulated upon MSC introduction. This suggests that MSC and dendritic cell interaction is secretome driven, which is not largely attenuated upon their mutual interaction. The innate and responsive secretome analysis of MSCs with LPS and IFNγ has identified that six secretory molecules (IL-6, RANTES, IP-10, VEGF, MIG, GCSF), which are upregulated in the cocultures are sourced from MSCs and are either innately secreted or upregulated upon stimulation.

Previous studies have shown that GCSF promotes the in vitro differentiation of human dendritic cells [49]. The roles of IL-6, RANTES, IP-10, VEGF, and MIG in modulating dendritic cell migration, differentiation, and other responses, have also been reported [50,51,52,53,54,55]. In addition to these secretory factors that are upregulated in the cocultures, our results also show that LPS and IFNγ upregulate other secretory factors on MSCs, albeit with differential magnitude and significance. Several of these secretory molecules also play a role in modulating dendritic cell function, notably MCP-1, since previous studies have shown that this plays a role in dendritic cell maturation and function [31,56]. However, it is difficult to propose that a single molecule/pathway exclusively determines the interactions of MSCs with dendritic cells. This is largely due to a variety of cytokines and growth factors that could possibly alter the maturation and responses of dendritic cells in an autocrine and paracrine fashion. Thus, our study provided a perspective that the cumulative and overlapping versatile secretory factors but not solely through a single effector mechanism, which plays a role in MSCs’ interaction with dendritic cells.

Our results show that MSCs do not secrete TNFα in their resting and activated status while potently attenuating its secretion from dendritic-cell-enriched PBMC cultures. Considering that TNFα is the potent inflammatory mediator in multiple clinical inflammatory and degenerative disorders [57], our results re-confirm the veto function of MSCs in suppressing inflammation and promoting immunomodulation. Our results also show that MSCs do not block the Type I interferon (IFNα) secretion from the TLR7-activated plasmocytoid dendritic cells. Considering this dichotomic functionality of MSCs in modulating secretory repertoire, future research is needed to identify the relative significance of these secretory modulations in achieving clinical benefit.

Our study investigated PBMCs instead of whole blood. This is largely due to the technical challenges associated with experimentation with fresh whole blood, such as feasibility, and the low frequency of circulating dendritic cell subsets in the peripheral blood to perform functional experiments. Nevertheless, future research is warranted to investigate MSC’s immunomodulatory effect on whole blood since emerging literature clearly indicates that therapeutic cells are targeted through the complement activation and coagulation cascade in the plasma/serum proteins in a process named Instant-Blood-Mediated Inflammatory Reaction (IBMIR) [3,58,59,60,61,62]. In addition, future investigations are also required to define the modulating effect of MSCs on circulating dendritic cell subsets from patients with inflammatory and degenerative disorders.

An important component of an efficacious MSC therapy is defining its potency as a cellular product that is infused into the patients [63]. Although the functional characteristics of MSCs are well studied, challenges exist in elucidating their potency. This is largely due to the evolving knowledge concerning the mechanisms of action of MSCs in achieving therapeutic benefits. It is being recognized that infused MSCs operate through multiple mechanisms, including macrophage polarization through efferocytosis process, the trans-differentiation mediated replacement of damaged tissues and instruction of immune cells, which, cumulatively or overlappingly, contributes to the clinical outcome [14]. Of these effector mechanisms, the instruction of immune cells resulting in systemic and/or local immunomodulation is of wide interest due to the fact that MSCs are generally being tested or used as a therapy for inflammatory and degenerative disorders. Our study informs the knowledge on MSCs’ interaction with circulating dendritic cells, which can be part of the potency metrics when evaluating the instruction of immune cells by MSCs. We identified a large secretome signature that predicts the interaction of MSCs with myeloid but not plasmacytoid dendritic cells. Our results provide clues that the interaction of MSC with circulating myeloid dendritic cells and the associated secretome signature could be a part of MSC’s potency evaluation. Hence, future MSC clinical trials need to investigate the relationship of circulating dendritic cell subsets and associated peripheral secretomes with clinical outcomes.

## 5. Conclusions

The present study showed contrariety of MSC functionality in modulating myeloid and plasmacytoid dendritic cells and the associated secretomes. The translational significance of this contrariety requires further investigation in clinical trials to test the application of MSC therapy as a potency biomarker.

## Figures and Tables

**Figure 1 biology-12-00725-f001:**
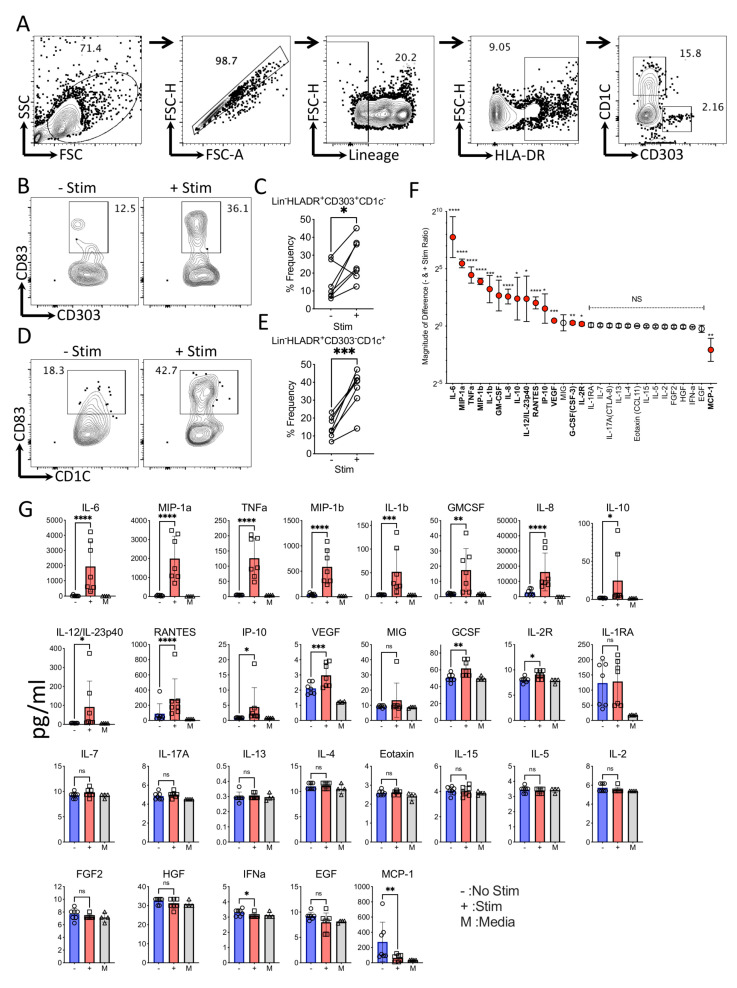
Characteristics of circulating dendritic cells with and without LPS and IFNγ stimulation. Dendritic-cell-enriched PBMCs were stimulated with LPS and IFNγ for 24 h. Subsequently, dendritic cell maturation was measured by flow cytometry and secretory analytes were quantified in the supernatants using Luminex xMAP (multi-analyte profiling) technology. (**A**) Representative flow cytometry plot and gating strategy are shown to identify plasmacytoid (Lineage−(CD3−CD14−CD16−CD56−CD19−CD20−) HLADR+CD303+CD1C−) and myeloid (Lineage−(CD3−CD14−CD16−CD56−CD19−CD20−) HLADR+CD303−CD1C+) dendritic cell subtypes. CD83 expression between−and + stim conditions are shown for (**B**,**C**) Lineage−HLADR+CD303+CD1C− and (**D**,**E**) Lineage−HLADR+CD303−CD1C+ populations with representative and cumulative plots. (**F**) Twnety-nine secretory analytes are organized based on their magnitude of differences between stim and no stim conditions. The magnitude of secretory difference is calculated as the geometric mean of the ratio between stim and no stim condition with 95% confidence interval. (**G**) concentrations of individual analytes are shown between stim (+) and no stim (−) cultures with mean and standard deviation. Statistical significance is shown as ns, *p* > 0.05; * *p* ≤ 0.05; ** *p* ≤ 0.01; *** *p* ≤ 0.001; **** *p* ≤ 0.0001. M indicates media control. Cumulative results are shown from seven independent donors.

**Figure 2 biology-12-00725-f002:**
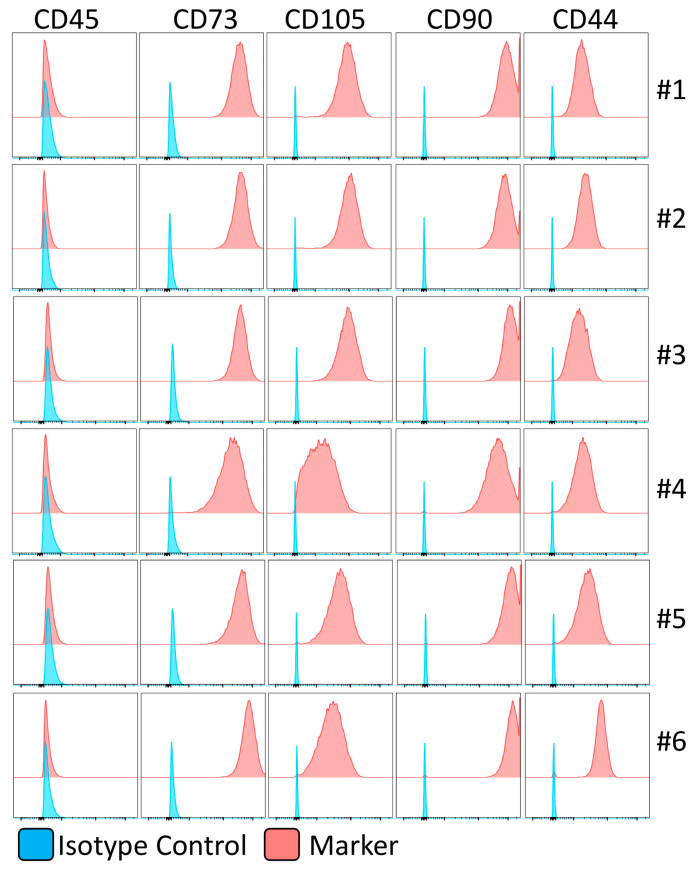
Immunophenotype of human bone-marrow-derived mesenchymal stromal cells. The identity of human bone-marrow-derived MSCs is confirmed in flow cytometry. Histogram plots are shown with appropriate isotype controls (**blue**) and marker (**red**) expressions. The results are shown from six independent donors.

**Figure 3 biology-12-00725-f003:**
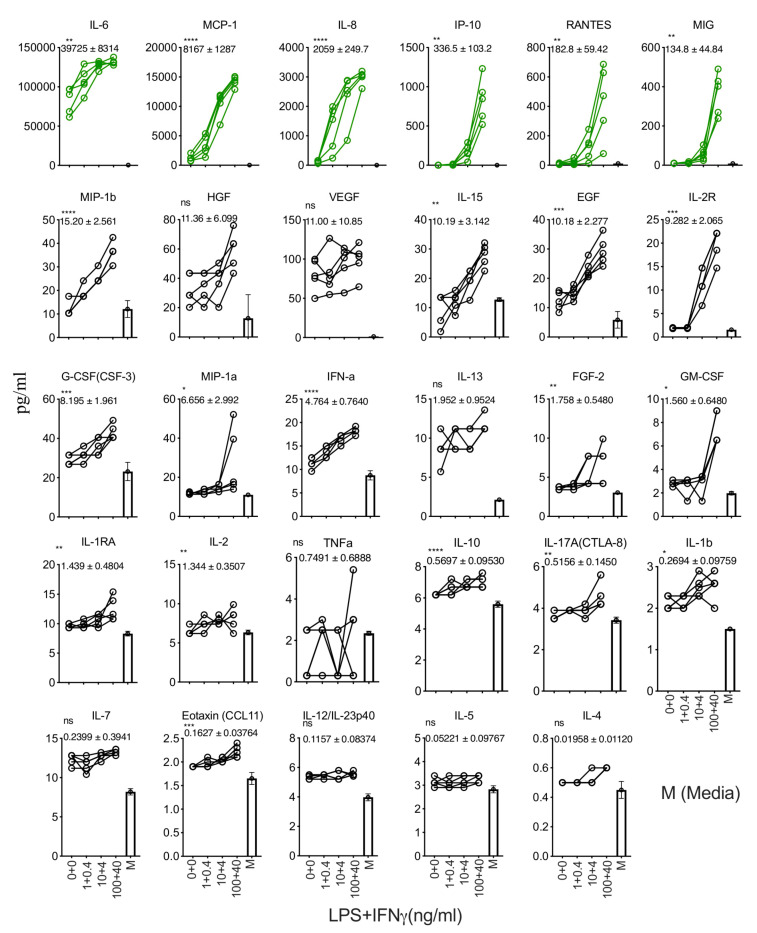
Dose-dependent effect of LPS and IFNγ stimulation of human bone marrow MSCs. MSCs derived from five independent donors were seeded at the identical cell density (10,000 cells/well in 96-well plate) and stimulated with the indicated concentrations of LPS and IFNγ. Forty-eight hours later, supernatants were centrifuged and stored. Secretory analytes were quantified in the supernatants using Luminex xMAP (multi-analyte profiling) technology. Twenty-nine analytes are hierarchically organized based on their magnitude of response to the stimulus LPS and IFNγ. The magnitude of upregulation and 95% confidence interval is shown for each analyte. Statistical significance is shown as ns, *p* > 0.05; * *p* ≤ 0.05; ** *p* ≤ 0.01; *** *p* ≤ 0.001; **** *p* ≤ 0.0001. M indicates media control. Cumulative results are shown from five independent donors.

**Figure 4 biology-12-00725-f004:**
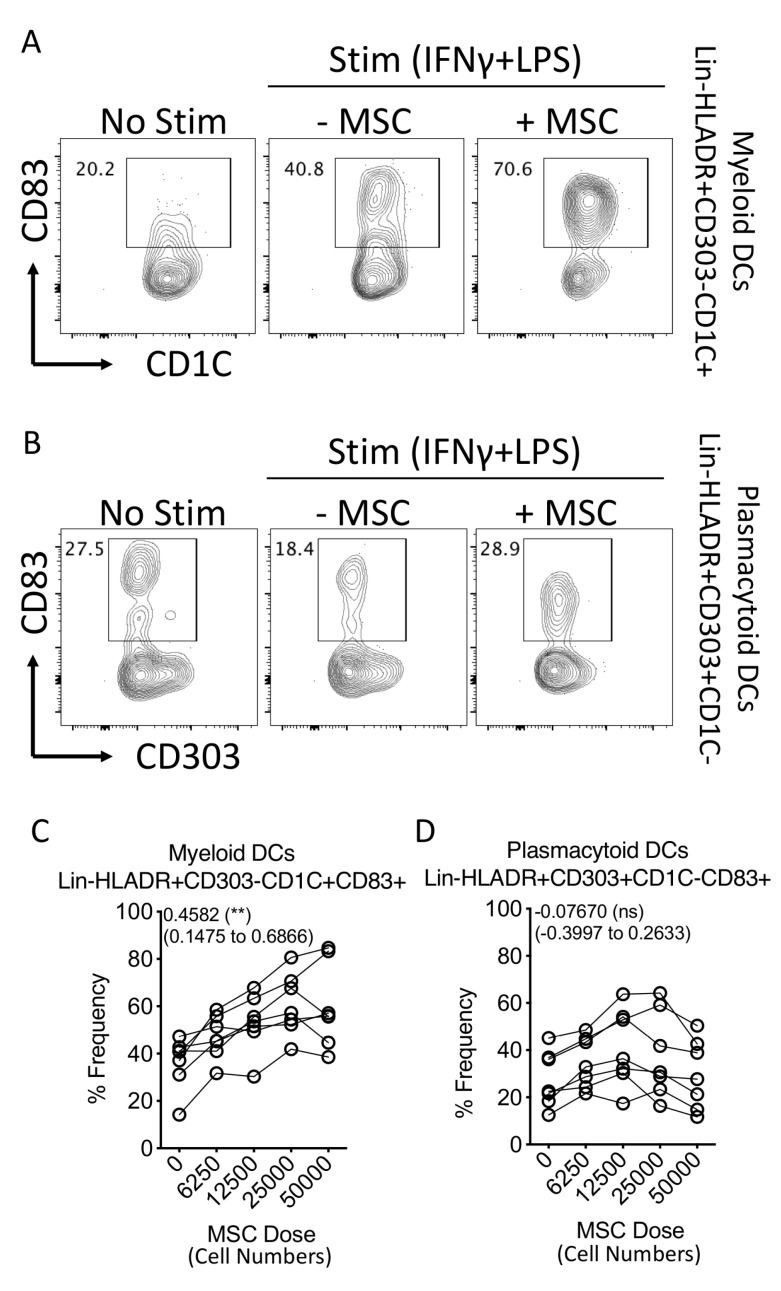
Effect of human bone marrow MSCs on the maturation of dendritic cell subsets. Human bone-marrow-derived MSCs were seeded at different cell densities. Subsequently, dendritic-cell-enriched PBMCs, stimulated with LPS (100 ng/mL) and IFNγ (40 ng/mL) were cocultured with MSCs. Twenty-four hours post incubation, they were stained with antibodies for lineage, HLADR, CD303, CD1C and CD83 and acquired in flow cytometer. Representative flow cytometry plots of (**A**) Lineage-(CD3−CD14−CD16−CD56−CD19−CD20−) HLADR+CD303−CD1C+ and (**B**) Lineage− (CD3−CD14−CD16−CD56−CD19−CD20−) HLADR+CD303+CD1C- are shown for the expression of CD83 with and without MSCs. Spaghetti plots depict the frequencies of (**C**) Lineage−HLADR+CD303−CD1C+CD83+ and (**D**) Lineage−HLADR+CD303+CD1C−CD83+ populations in the presence of various doses of MSCs. Dose dependence values, statistical significance and 95% confidence interval are shown in individual plots. Cumulative data derived from six independent MSC and seven PBMC donors are shown.

**Figure 5 biology-12-00725-f005:**
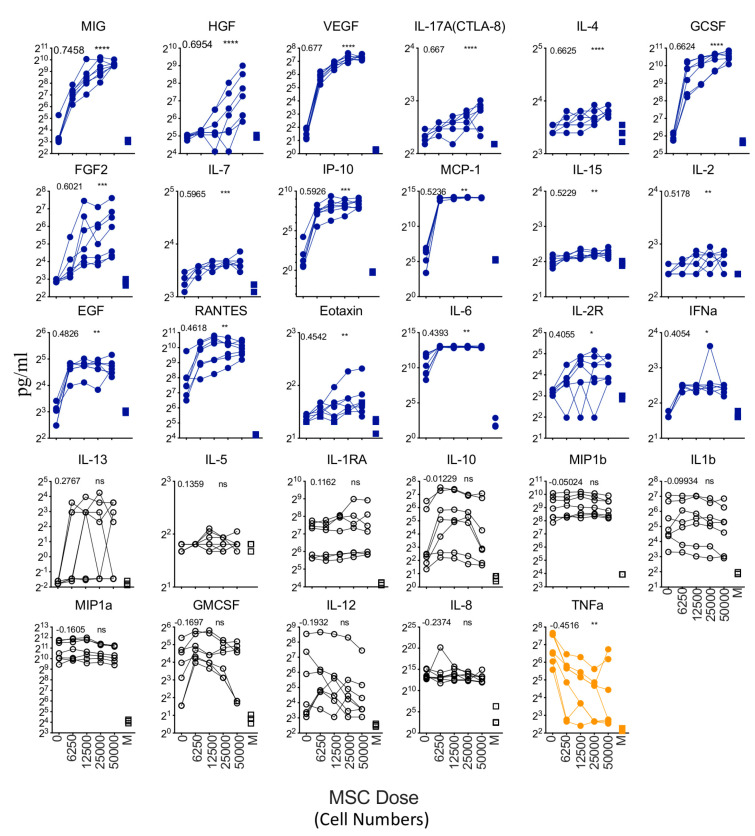
Dose-dependent effect of MSCs in modulating the secretome of LPS- and IFNγ-activated DC enriched PBMCs. Supernatants derived from the cocultures of LPS and IFNγ stimulated dendritic-cell-enriched PBMCs and MSCs were analyzed for 30-plex secretome using Luminex xMAP (multi-analyte profiling) technology. Spaghetti plots depict the levels of individual secretory molecules (pg/mL) in MSC dose-dependent cocultures with dendritic-cell-enriched PBMCs. Statistically significant secretory molecules that are upregulated with the increasing doses of MSCs in the cocultures are highlighted in blue. In contrast, statistically significant secretory molecules that are downregulated with the increasing doses of MSCs in the cocultures are highlighted in orange. Statistically non-significant molecules are depicted in black. M stands for Media only control (No MSCs and No DC-enriched PBMCs). Dose dependency values and statistical significance are shown in each graph (ns, *p* > 0.05; * *p* ≤ 0.05; ** *p* ≤ 0.01; *** *p* ≤ 0.001; **** *p* ≤ 0.0001). IFNγ in the 30-plex is excluded in the analysis since it is present in the inoculum. Cumulative data derived from six independent MSC and seven PBMC donors are shown.

**Figure 6 biology-12-00725-f006:**
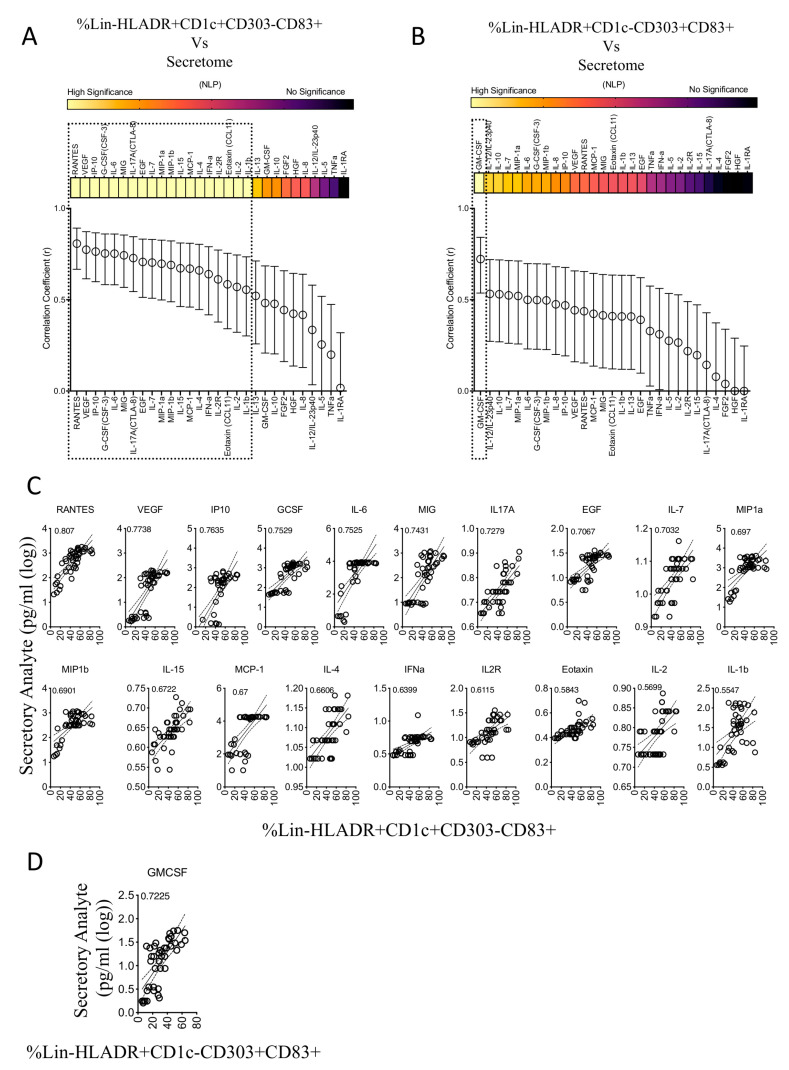
Correlation patterns between MSC-induced frequency of dendritic cell subsets and secretome. Quantitative levels of individual secretory molecules derived from the cocultures of MSCs and dendritic-cell-enriched PBMCs were subjected to linear regression analysis with the corresponding percentage of (**A**) Lineage-HLADR+CD303-CD1C+CD83+ and (**B**) Lineage-HLADR+CD303+CD1C-CD83+ populations. Correlation coefficient r-values of 1 and 0 specify the best direct and no correlations, respectively. The statistical significance of each of these correlations is shown with the heat map that describes the negative logarithmic of *p*-values. Concentration secretory analytes were transformed to logarithmic values to fit linear regression curves. Statistically high significant correlations with (**C**) Lineage-HLADR+CD303-CD1C+CD83+ and (**D**) Lineage-HLADR+CD303+CD1C-CD83+ populations are shown in the individual plots. Cumulative data derived from six independent MSC and seven PBMC donors are used in the linear regression analysis.

**Figure 7 biology-12-00725-f007:**
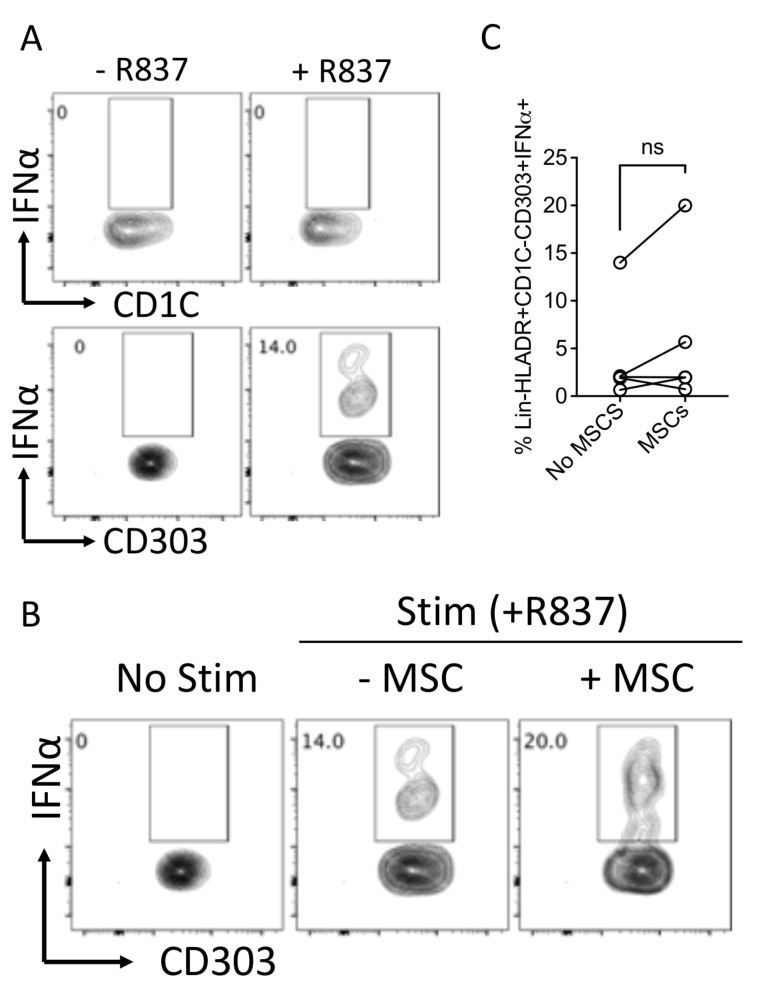
Effect of human bone-marrow-derived MSCs in modulating IFNα production in plasmacytoid dendritic cells. PBMCs (100,000 cells/well) were cocultured in the presence and absence of MSCs (10,000 cells/well) and were stimulated with R837 (20 μg/mL). The cells were incubated with BD Golgi Plug for 14 h and subsequently stained with surface antibodies for Lineage, HLADR, CD303, CD1C and intracellular IFNα. (**A**) Representative flow plots describing the IFNα production in Lineage-HLADR+CD303-CD1C+ and Lineage-HLADR+CD303+CD1C- populations are shown. (**B**) The effect of MSCs in the frequency of Lineage-HLADR+CD303+CD1C- IFNα+ populations are shown with a representative flow plot and (**C**) cumulative graphs. (**C**) Cumulative data derived from five independent MSC and five PBMC donors are shown. Statistical significance is shown. ns = Nonsignificant.

## Data Availability

Data is contained within the article.

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
