# Peer review of "Contrariety of Human Bone Marrow Mesenchymal Stromal Cell Functionality in Modulating Circulatory Myeloid and Plasmacytoid Dendritic Cell Subsets"

_biology, 2023, doi:10.3390/biology12050725_

Round 1
Reviewer 1 Report
Dear authors
I've find some minor issues during the paper that are listed below. In addition, I missed some functional studies as dendritic cell function evaluation as an antigen presenting cell.
- Line 23 is missing a comma after "overall"
- Line 86, 93 seems to have double space
- Line 118 - Add gate strategy for the flow cytometry analysis
- All figures are with a "contour box" , I suggest take it out as the labeling on the bottom of right side
- Line 180 - Something is wrong with Figure 1 legend, Line 181 is in bold and that is no connection with line 180.
- y axis from cytokine analysis unit must be pg/ml not Pg/ml
- p values is in lower case not upper as it is in legends.
- Figure 4 and Figure 5 miss the unit for x axis
- Line 252, 295, 304 have double space
- Figure 6 also miss the unit for y axis
- "Figure 7" is not bold
- References are not is the same letter format (font and size)
Reviewer 2 Report
Dear authors, the analytical part of this study is relevant and technically well executed and these data are of interest to the scientific community working with MSC therapeutics. However, there are some minor shortcomings considering the description of the actual study setup that must be improved, so that the study becomes more understandable for the readers. I would suggest Minor Revision
1) Some sections are somewhat misleading in the current format, but this can be easily rectified, e.g. it is not clear to me if this is an analytical follow-up of an actual clinical study based on analyzing patient-derived PBMCs/DC-subsets, or if this is an experimental in vitro setup. This is currently not clearly expressed in the paper. Either way, the experimental / study setup needs to be properly described in the methods section and graphical abstract to better introduce the study. Even if it is an in vitro coculture study, the authors should clearly indicate how this was done.
2) To improve the impact/relevance of the study, it would be of interest to not only study cocultures of MSCs with isolated DCs, but to also do this FACS readout in a whole-blood setup, both with healthy donor and patient blood (e.g. hirudin-anticoagulated blood to maintain complement activity allows cocultures of MSCs with whole blood for many hours, followed by erythrocyte lysis and fixation the same advanced FACS panel could be used for readout, and similar cytokine analysis performed on the blood serum/plasma), to better replicate the situation in clinical trials, e.g. see PMID 21747949 PLOS1 2011 “Mesenchymal Stromal Cells Engage Complement and Complement Receptor Bearing Innate Effector Cells to Modulate Immune Responses”. Importantly, complement activation by infused MSCs modulates the outcome of their interaction with myeloid effector cells (please see graphical summary).
3) The authors describe in their abstract and introduction clinical trials on MSC infusion, and how the infused therapeutic MSCs interact with the host immune system, e.g. different subsets of lymphoid or myeloid cells, such as monocytes and different dendritic cell (DC) subsets (e.g. myeloid and plasmacytoid DCs). The authors then state that more detailed knowledge on the effect of infused MSCs on circulating DC subsets is still lacking, e.g. shifts in respective frequency and activation/cytokine production. The authors then summarize their main results, that MSCs promote the maturation of myeloid DCs, but that they do not significantly affect the response of plasmacytoid DCs. The authors then conduct some mechanistic in vitro follow-up. The methods section lists the manufacturing of BM-MSCs (2.1), PBMC isolation and DC enrichment from healthy donors and cocultures with MSCs (2.2.), flow cytometry and multiplex analysis (2.3 and 2.4) and statistics (2.5). However, I can’t find any information on clinical trials of MSC infusion and PBMC collection, as the abstract and introduction section may imply that this is a follow-up of a trial, e.g. results section 3.1 speaks again on the effect of MSCs on circulating DC subsets. The authors clearly need to separate in vivo from in vitro results, since there might be fundamental differences in the mechanisms of action. This would greatly improve the quality of the paper. The authors should compose a graphical abstract that clearly describes the principal study setup and the major results.
4) Importantly, the following results sections describe several mechanistic experiments on MSC interaction with healthy donor PBMC-derived DCs in different coculture setups, which does not really approximate the interaction of infused therapeutic MSCs with circulating PBMCs/DCs in the clinical patient whole blood environment for two specific reasons: 1) Human blood serum/plasma proteins are missing in this setup, such as complement/coagulation/fibrinolytic cascades (please see comments above), and 2) Healthy donor blood was used as source in this study, which differs in its composition from patient blood, different types of cytokines profiles and immune effector cells and respective activation status. These points should be discussed in the Discussion section. For further reading on this point please see the following original references and review articles where this is clearly introduced and discussed, e.g. how therapeutic cells are targeted by complement activation and coagulation collectively termed instant-blood-mediated inflammatory reaction (IBMIR): PMID 22522999 Stem Cells 2012 “Are therapeutic human mesenchymal stromal cells compatible with human blood?”, PMID 24805247 Stem Cells 2014 “Do cryopreserved mesenchymal stromal cells display impaired immunomodulatory and therapeutic properties?”, PMID: 27837556 “Cryopreserved or Fresh Mesenchymal Stromal Cells: Only a Matter of Taste or Key to Unleash the Full Clinical Potential of MSC Therapy?”, PMID 30711482 Trends in Molecular Medicine 2019 ”Intravascular MSC Therapy Product Diversification: Time for New Clinical Guidelines”, and PMID 35641163 Stem Cells Translational Medicine 2022 “Improved MSC minimal criteria to maximize patient safety: a call to embrace tissue factor and hemocompatibility assessment of MSC products.”
Reviewer 3 Report
In the manuscript entitled “Dynamics of Human Bone Marrow Stromal Cell Interaction with Circulating Dendritic Cell Subsets”, this paper is informative to understand the mechanism of MSCs with circulating DC subsets. I think that this paper about MSCs therapy will be read by many researchers in this field and cited in future. Therefore, I recommend that this paper is acceptable for publication in biology after minor point as below.
- Please change the submitted title to specific title including experimental results.
